# Finding ReMO (Related Memory Object): A Simple Neural Architecture for Text based Reasoning

## Abstract

To solve the text-based question and answering task that requires relational reasoning, it is necessary to memorize a large amount of information and find out the question relevant information from the memory. Most approaches were based on external memory and four components proposed by Memory Network. The distinctive component among them was the way of finding the necessary information and it contributes to the performance. Recently, a simple but powerful neural network module for reasoning called Relation Network (RN) has been introduced. We analyzed RN from the view of Memory Network, and realized that its MLP component is able to reveal the complicate relation between question and object pair. Motivated from it, we introduce *Relation Memory Network (RMN)* which uses MLP to find out relevant information on Memory Network architecture. It shows new state-of-the-art results in jointly trained bAbI-10k story-based question answering tasks and bAbI dialog-based question answering tasks.

## 1 Introduction

Neural network has made an enormous progress on the two major challenges in artificial intelligence: seeing and reading. In both areas, embedding methods have served as the main vehicle to process and analyze text and image data for solving classification problems. As for the task of logical reasoning, however, more complex and careful handling of features is called for. A reasoning task requires the machine to answer a simple question upon the delivery of a series of sequential information. For example, imagine that the machine is given the following three sentences: "Mary got the milk there.", "John moved to the bedroom.", and "Mary traveled to the hallway." Once prompted with the question, "Where is the milk?", the machine then needs to sequentially focus on the two supporting sentences, "Mary got the milk there." and "Mary traveled to the hallway." in order to successfully determine that the milk is located in the hallway.

Inspired by this reasoning mechanism, J. Weston & Bordes (2015) has introduced the memory network (MemNN), which consists of an external memory and four components: input feature map ($I$), generalization ($G$), output feature map ($O$), and response ($R$). The external memory enables the model to deal with a knowledge base without loss of information. Input feature map embeds the incoming sentences. Generalization updates old memories given the new input and output feature map finds relevant information from the memory. Finally, response produces the final output.

Based on the memory network architecture, neural network based models like end-to-end memory network (MemN2N) (Sukhbaatar et al., 2015), gated end-to-end memory network (GMemN2N) (Liu & Perez, 2017), dynamic memory network (DMN) (Kumar et al., 2016), and dynamic memory network + (DMN+) (Xiong et al., 2016) are proposed. Since strong reasoning ability depends on whether the model is able to sequentially catching the right supporting sentences that lead to the answer, the most important thing that discriminates those models is the way of constructing the output feature map. As the output feature map becomes more complex, it is able to learn patterns for more complicate relations. For example, MemN2N, which has the lowest performance among the four models, measures the relatedness between question and sentence by the inner product, while the best performing DMN+ uses inner product and absolute difference with two embedding matrices.

Recently, a new architecture called Relation Network (RN) (Santoro et al., 2017) has been proposed as a general solution to relational reasoning. The design philosophy behind it is to directly capture the supporting relation between the sentences through the multi-layer perceptron (MLP). Despite its simplicity, RN achieves better performance than previous models without any catastrophic failure.

The interesting thing we found is that RN can also be interpreted in terms of MemNN. It is composed of $O$ and $R$ where each corresponds to MLP which focuses on the related pair and another MLP which infers the answer. RN does not need to have $G$ because it directly finds all the supporting sentences at once. In this point of view, the significant component would be MLP-based output feature map. As MLP is enough to recognize highly non-linear pattern, RN could find the proper relation better than previous models to answer the given question.

However, as RN considers a pair at a time unlike MemNN, the number of relations that RN learns is $n^2$ when the number of input sentence is $n$. When $n$ is small, the cost of learning relation is reduced by $n$ times compared to MemNN based models, which enables more data-efficient learning (Santoro et al., 2017). However, when $n$ increases, the performance becomes worse than the previous models. In this case, the pair-wise operation increases the number of non-related sentence pairs more than the related sentence pair, thereby confuses RN's learning. Santoro et al. (2017) has suggested attention mechanisms as a solution to filter out unimportant relations; however, since it interrupts the reasoning operation, it may not be the most optimal solution to the problem.

Our proposed model, "Relation Memory Network" (RMN), is able to find complex relation even when a lot of information is given. It uses MLP to find out relevant information with a new generalization which simply erase the information already used. In other words, RMN inherits RN's MLP-based output feature map on Memory Network architecture. Experiments show its state-of-the-art result on the text-based question answering tasks.

## 2  RELATION MEMORY NETWORK

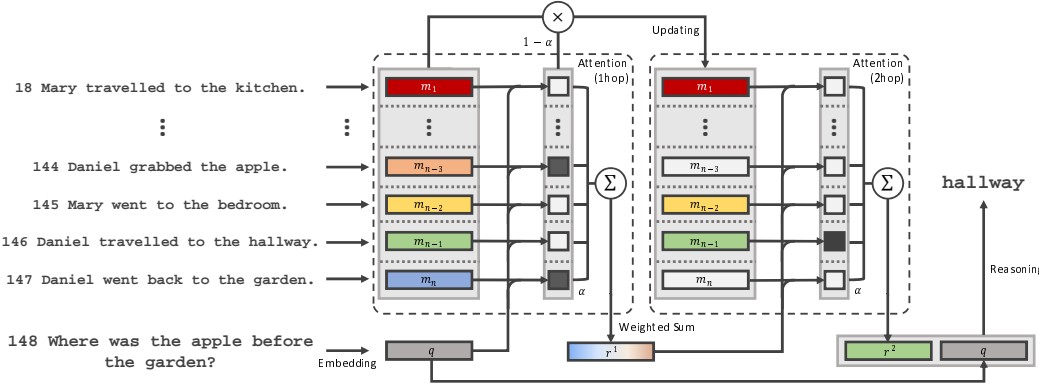

Figure 1: Relation Memory Network

Relation Memory Network (RMN) is composed of four components - embedding, attention, updating, and reasoning. It takes as the inputs a set of sentences $\mathbf{x}_1, \mathbf{x}_2, ..., \mathbf{x}_n$ and its related question $\mathbf{u}$, and outputs an answer $\mathbf{a}$. Each of the $\mathbf{x}_i$, $\mathbf{u}$, and $\mathbf{a}$ is made up of one-hot representation of words, for example, $\mathbf{x}_i = \{\mathbf{x}_{i1}, \mathbf{x}_{i2}, \mathbf{x}_{i3}, ..., \mathbf{x}_{in_i}\}$ $(\mathbf{x}_{ij} \in \mathbb{R}^V, j = (1, 2, ..., n_i), V = $ vocabulary size, $n_i = $ number of words in sentence $i$).

### 2.1  EMBEDDING COMPONENT

We first embed words in each $\mathbf{x}_i = \{\mathbf{x}_{i1}, \mathbf{x}_{i2}, \mathbf{x}_{i3}, ..., \mathbf{x}_{in_i}\}$ and $\mathbf{u}$ to a continuous space multiplying an embedding matrix $A \in \mathbb{R}^{d \times V}$. Then, the embedded sentence is stored and represented as a memory object $\mathbf{m}_i$ while question is represented as $\mathbf{q}$. Any of the following methods are available for embedding component: simple sum (equation 1), position encoding (J. Weston & Bordes, 2015) (equation 2), concatenation (equation 3), LSTM, and GRU. In case of LSTM or GRU, $\mathbf{m}_i$ is the

final hidden state of it.

$$\mathbf{m}_i = \sum_j A\mathbf{x}_{ij} \tag{1}$$

$$\mathbf{m}_i = \sum_j l_j \cdot A\mathbf{x}_{ij} \quad (l_{kj} = (1 - j/n_i) - (k/d)(1 - 2j/n_i)) \tag{2}$$

$$\mathbf{m}_i = [A\mathbf{x}_{i1}, A\mathbf{x}_{i2}, ... , A\mathbf{x}_{in_i}] \tag{3}$$

As the following attention component takes the concatenation of $\mathbf{m}_i$ and $\mathbf{q}$, it is not necessarily the case that sentence and question have the same dimensional embedding vectors unlike previous memory-augmented neural networks.

## 2.2 ATTENTION COMPONENT

Attention component can be applied more than once depending on the problem; Figure 1 illustrates 2 hop version of RMN. We refer to the $i^{th}$ embedded sentence on the $t^{th}$ hop as $\mathbf{m}_i^t$.

To constitute the attention component, we applied simple MLP represented as $g_\theta^t$. It must be ended with 1 unit output layer to provide a scalar weight $w_i^t$, which leads to an attention weight $\alpha_i^t$ between 0 and 1. In the beginning, a vector concatenated with $\mathbf{m}_i^1$ and $\mathbf{q}$ flows to the $g_\theta^1$. From the result of $g_\theta^1$, attention weight $\alpha_i^1$ is calculated using additional variable $\beta^1$ ($\geq 1$) to control the intensity of attention, inspired by the way Neural Turing Machine (Graves et al., 2014) reads from the memory. Then we get the related memory object $\mathbf{r}^1$, a weighted sum of $\alpha_i^1$ and memory object $\mathbf{m}_i^1$ for all $i$. If there exist more hops, $\mathbf{r}^1$ is directly taken to the next hop and iterates over this process with the updated memory object $\mathbf{m}_i^2$. All the procedures are rewritten as equation 4, 5, and 6:

$$w_i^t \leftarrow g_\theta^t([\mathbf{m}_i^t, \mathbf{r}^{t-1}]) \quad (i = (1, 2, ..., n),\ \mathbf{r}^0 = \mathbf{q}) \tag{4}$$

$$\alpha_i^t \leftarrow \frac{\exp(\beta^t w_i^t)}{\sum_i \exp(\beta^t w_i^t)} \quad (\beta^t(z) = 1 + \log(1 + \exp(z))) \tag{5}$$

$$\mathbf{r}^t \leftarrow \sum_i \alpha_i^t \cdot \mathbf{m}_i^t \tag{6}$$

## 2.3 UPDATING COMPONENT

To forget the information already used, we use intuitive updating component to renew the memory. It is replaced by the amount of unconsumed from the old one:

$$\mathbf{m}_i^{t+1} \leftarrow (1 - \alpha_i^t)\,\mathbf{m}_i^t \tag{7}$$

Contrary to other components, updating is not a mandatory component. When it is considered to have 1 hop, there is no need to use this.

## 2.4 REASONING COMPONENT

Similar to attention component, reasoning component is also made up of MLP, represented as $f_\phi$. It receives both $\mathbf{q}$ and the final result of attention component $\mathbf{r}^f$ and then takes a softmax to produce the model answer $\hat{\mathbf{a}}$:

$$\hat{\mathbf{a}} \leftarrow \text{Softmax}(f_\phi([\mathbf{r}^f, \mathbf{q}])) \tag{8}$$

## 3 RELATED WORK

### 3.1 MEMORY-AUGMENTED NEURAL NETWORK

To answer the question from a given set of facts, the model needs to memorize these facts from the past. Long short term memory (LSTM) (Hochreiter & Schmidhuber, 1997), one of the variants of recurrent neural network (RNN), is inept at remembering past stories because of their small internal memory (Sukhbaatar et al., 2015). To cope with this problem, J. Weston & Bordes (2015) has

Table 1: MemN2N, RN, and RMN in terms of MemNN architecture

| | Output feature map | Generalization |
|---|---|---|
| MemN2N | $\alpha_i^t = \text{Softmax}((\mathbf{r}^t)^T \mathbf{m}_i^t)\,(\mathbf{r}^0 = \mathbf{q})$ 
 $\mathbf{o}^t = \sum_i \alpha_i^t \mathbf{m}_i^t$ 
 $\mathbf{r}^{t+1} = \mathbf{o}^t + \mathbf{r}^t$ | $\mathbf{m}^t = W^t \mathbf{m}^{t-1}$ |
| RN | $\mathbf{r}_i = g_\theta\text{-MLP}([\mathbf{m}_i, \mathbf{m}_j, \mathbf{q}])$ 
 $\mathbf{o} = \sum_i \mathbf{r}_i$ | - |
| RMN | $\mathbf{r}_i^t = g_\theta^t\text{-MLP}([\mathbf{m}_i^t, \mathbf{r}^{t-1}])\,(\mathbf{r}^0 = \mathbf{q})$ 
 $\alpha_i^t = \text{Softmax}(\beta^t \mathbf{r}_i^t)$ 
 $\mathbf{r}^t = \sum_i \alpha_i^t \mathbf{m}_i^t$ | $\mathbf{m}_i^t = (1 - \alpha_i^{t-1})\mathbf{m}_i^{t-1}$ |

proposed a new class of memory-augmented model called Memory Network (MemNN). MemNN comprises an external memory $m$ and four components: input feature map ($I$), generalization ($G$), output feature map ($O$), and response ($R$). $I$ encodes the sentences which are stored in memory $m$. $G$ updates the memory, whereas $O$ reads output feature $o$ from the memory. Finally, $R$ infers an answer from $o$.

MemN2N, GMemN2N, DMN, and DMN+ all follow the same structure of MemNN from a broad perspective, however, output feature map is composed in slightly different way. The relation between question and supporting sentences is realized from its cooperation. MemN2N first calculates the relatedness of sentences in the question and memory by taking the inner product, and the sentence with the highest relatedness is selected as the first supporting sentence for the given question. The first supporting sentence is then added with the question and repeat the same operation with the updated memory to find the second supporting sentence. GMemN2N selects the supporting sentence in the same way as MemN2N, but uses the gate to selectively add the the question to control the influence of the question information in finding the supporting sentence in the next step. DMN and DMN + use output feature map based on various relatedness such as absolute difference, as well as inner product, to understand the relation between sentence and question at various points.

The more difficult the task, the more complex the output feature map and the generalization component to get the correct answer. For a dataset experimenting the text-based reasoning ability of the model, the overall accuracy could be increased in order of MemN2N, GMemN2N, DMN, and DMN+, where the complexity of the component increases.

## 3.2 RELATION NETWORK

Relation Network (RN) has emerged as a new and simpler framework for solving the general reasoning problem. RN takes in a pair of objects as its input and simply learns from the compositions of two MLPs represented as $g_\theta$ and $f_\phi$. The role of each MLP is not clearly defined in the original paper, but from the view of MemNN, it can be understood that $g_\theta$ corresponds to $O$ and $f_\phi$ corresponds to $R$. Table 1 summarizes the interpretation of RN compared to MemN2N and our model, RMN.

To verify the role of $g_\theta$, we compare the output when pairs are made with supporting sentences and when made with unrelated sentences. Figure 2 shows the visualization result of each output. When we focus on whether the value is activated or not, we can see that $g_\theta$ distinguishes supporting sentence pair from non-supporting sentence pair as output feature map examines how relevant the sentence is to the question. Therefore, we can comprehend the output of $g_\theta$ reveals the relation between the object pair and the question and $f_\phi$ aggregates all these outputs to infer the answer.

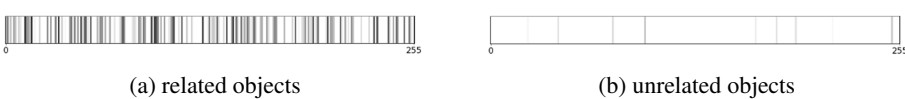

(a) related objects                    (b) unrelated objects

Figure 2: The output vector of $g_\theta$ when the input objects are related and unrelated

# 4 EXPERIMENTS

## 4.1 DATASET

**bAbI story-based QA dataset**    bAbI story-based QA dataset (Weston et al., 2015) is composed of 20 different types of tasks for testing natural language reasoning ability. Each task requires different methods to infer the answer. The dataset includes a set of statements comprised of multiple sentences, a question and answer. A statement can be as short as two sentences and as long as 320 sentences. To answer the question, it is necessary to find relevant one or more sentences to a given question and derive answer from them. Answer is typically a single word but in a few tasks, answers are a set of words. Each task is regarded as success when the accuracy is greater than 95%. There are two versions of this dataset, one that has 1k training examples and the other with 10k examples. Most of the previous models test their accuracy on 10k dataset with trained jointly.

**bAbI dialog dataset**    bAbI dialog dataset (Bordes & Weston, 2016) is a set of 5 tasks within the goal-oriented context of restaurant reservation. It is designed to test if model can learn various abilities such as performing dialog management, querying knowledge bases (KBs), and interpreting the output of such queries. The KB can be queried using API calls and 4 fields (a type of cuisine, a location, a price range, and a party size). They should be filled to issue an API call. Task 1 tests the capacity of interpreting a request and asking the right questions to issue an API call. Task 2 checks the ability to modify an API call. Task 3 and 4 test the capacity of using outputs from an API call to propose options in the order of rating and to provide extra-information of what user asks for. Task 5 combines everything. The maximum length of the dialog for each task is different: 14 for task 1, 20 for task 2, 78 for task 3, 13 for task 4, and 96 for task 5. As restaurant name, locations, and cuisine types always face new entities, there are normal and OOV test sets to assess model's generalization ability. Training sets consist fo 1k examples, which is not a large amount of creating realistic learning conditions.

## 4.2 TRAINING DETAILS

**bAbI story-based QA dataset**    We trained 2 hop RMN jointly on all tasks using 10k dataset for model to infer the solution suited to each type of tasks. We limited the input to the last 70 stories for all tasks except task 3 for which we limited input to the last 130 stories, similar to Xiong et al. (2016) which is the hardest condition among previous models. Then, we labeled each sentence with its relative position. Embedding component is similar to Santoro et al. (2017), where story and question are embedded through different LSTMs; 32 unit word-lookup embeddings; 32 unit LSTM for story and question. For attention component, as we use 2 hop RMN, there are $g_\theta^1$ and $g_\theta^2$; both are three-layer MLP consisting of 256, 128, 1 unit with ReLU activation function (Nair & Hinton, 2010). $f_\phi$ is composed of 512, 512, and 159 units (the number of words appearing in bAbI dataset is 159) of three-layer MLP with ReLU non-linearities where the final layer was a linear that produced logits for a softmax over the answer vocabulary. For regularization, we use batch normalization (Ioffe & Szegedy, 2015) for all MLPs. The softmax output was optimized with a cross-entropy loss function using the Adam optimizer (Kingma & Ba, 2014) with a learning rate of $2e^{-4}$.

**bAbI dialog dataset**    We trained on full dialog scripts with every model response as answer, all previous dialog history as sentences to be memorized, and the last user utterance as question. Model selects the most probable response from 4,212 candidates which are ranked from a set of all bot utterances appearing in training, validation and test sets (plain and OOV) for all tasks combined. We also report results when we use match type features for dialog. Match type feature is an additional label on the candidates indicating if word is found on the dialog history. For example, if the world 'Seoul' is found, then the 'location' field is checked to hint model this word is important and should be used in API call. This feature can alleviate OOV problem. Training was done with Adam optimizer and a learning rate of $1e^{-4}$ for all tasks. Additional model details are given in Appendix A.

Table 2: Test error on bAbI story-based tasks with 10k training samples

| Task | MemNN | MemN2N | GMemN2N | DMN | DMN+ | DNC | EntNet[1] | RN[2] | RMN |
|------|-------|--------|---------|-----|------|-----|---------|-----|-----|
| 1: Single Supporting Fact | **0.0** | **0.0** | **0.0** | **0.0** | **0.0** | **0.0** | 0.1 | **0.0** | **0.0** |
| 2: Two Supporting Facts | **0.0** | **0.3** | **0.0** | 1.8 | 0.3 | 0.4 | 2.8 | 8.3 | 0.5 |
| 3: Three Supporting Facts | **0.0** | 9.3 | **4.5** | 4.8 | 1.1 | 1.8 | 10.6 | 17.1 | 14.7 |
| 4: Two Argument Relations | **0.0** | **0.0** | **0.0** | **0.0** | **0.0** | **0.0** | **0.0** | **0.0** | **0.0** |
| 5: Three Argument Relations | 2.0 | 0.6 | 0.2 | 0.7 | 0.5 | 0.8 | 0.4 | 0.7 | 0.4 |
| 6: Yes/No Questions | **0.0** | **0.0** | **0.0** | **0.0** | **0.0** | **0.0** | 0.3 | **0.0** | **0.0** |
| 7: Counting | 15.0 | **3.7** | 1.8 | 3.1 | 2.4 | 0.6 | 0.8 | 0.4 | 0.5 |
| 8: Lists/Sets | 9.0 | 0.8 | 0.3 | 3.5 | 0.0 | 0.3 | 0.1 | 0.3 | 0.3 |
| 9: Simple Negation | **0.0** | 0.8 | **0.0** | **0.0** | **0.0** | 0.2 | **0.0** | **0.0** | **0.0** |
| 10: Indefinite Knowledge | 2.0 | 2.4 | 0.2 | 2.5 | 0.0 | 0.2 | 0.0 | 0.0 | 0.0 |
| 11: Basic Coreference | **0.0** | **0.0** | **0.0** | 0.1 | **0.0** | **0.0** | **0.0** | 0.4 | 0.5 |
| 12: Conjunction | **0.0** | **0.0** | **0.0** | **0.0** | **0.0** | **0.0** | **0.0** | **0.0** | **0.0** |
| 13: Compound Coreference | **0.0** | **0.0** | **0.0** | 0.2 | **0.0** | 0.1 | **0.0** | **0.0** | **0.0** |
| 14: Time Reasoning | 1.0 | **0.0** | **0.0** | **0.0** | **0.0** | 0.4 | 3.6 | **0.0** | **0.0** |
| 15: Basic Deduction | **0.0** | **0.0** | **0.0** | **0.0** | **0.0** | **0.0** | **0.0** | **0.0** | **0.0** |
| 16: Basic Induction | **0.0** | 0.4 | **0.0** | 0.6 | 45.3 | 33.1 | 52.1 | 4.9 | 0.9 |
| 17: Positional Reasoning | 35.0 | 40.7 | 27.8 | 40.4 | 4.2 | 12.0 | 11.7 | 1.6 | 0.3 |
| 18: Size Reasoning | **5.0** | 6.7 | 8.5 | **4.7** | 2.1 | 0.8 | 2.1 | 2.1 | 2.3 |
| 19: Path Finding | 64.0 | 66.5 | 31.0 | 65.5 | 0.0 | 3.9 | 63.0 | 3.2 | 2.9 |
| 20: Agent's Motivations | **0.0** | **0.0** | **0.0** | **0.0** | **0.0** | **0.0** | **0.0** | **0.0** | **0.0** |
| Mean error (%) | 6.7 | 6.6 | 3.7 | 6.4 | 2.8 | 2.7 | 7.4 | 2.0 | **1.2** |
| Failed tasks (err. >5%) | 4 | 4 | 3 | 2 | **1** | 2 | 4 | 2 | **1** |

Table 3: bAbI story-based task visualization of $\alpha$

(a) Task 7

| Seq. | **Task 7: Counting** | $\alpha^1$ | $\alpha^2$ |
|------|----------------------|-----------|-----------|
| 8 | John grabbed the apple there . | 0.02 | 0.00 |
| 9 | John gave the apple to Mary . | 0.08 | 0.16 |
| 10 | Mary passed the apple to John . | 0.17 | 0.24 |
| 11 | Mary journeyed to the hallway . | 0.00 | 0.01 |
| 13 | Sandra went to the garden . | 0.00 | 0.00 |
| 14 | Mary went to the kitchen . | 0.00 | 0.03 |
| 15 | Mary picked up the football there . | 0.29 | 0.24 |
| 16 | Mary picked up the milk there . | 0.27 | 0.32 |
| User input | How many objects is Mary carrying? | | |
| Answer | Two | | |
| Model answer | Two | [Correct] | |

(b) Task 14

| Seq. | **Task 14: Time reasoning** | $\alpha^1$ | $\alpha^2$ |
|------|------------------------------|-----------|-----------|
| 1 | Mary went back to the school yesterday . | 0.00 | 0.01 |
| 2 | Fred went to the school yesterday . | 0.00 | 0.00 |
| 3 | Julie went back to the kitchen yesterday . | 0.13 | 0.98 |
| 4 | Fred journeyed to the kitchen this morning . | 0.00 | 0.00 |
| 5 | This morning julie journeyed to the school . | 0.66 | 0.02 |
| 6 | This evening mary went back to the school . | 0.01 | 0.00 |
| 7 | This afternoon julie went to the bedroom . | 0.06 | 0.00 |
| User input | Where was Julie before the school ? | | |
| Answer | Kitchen | | |
| Model answer | Kitchen | [Correct] | |

(c) Task 3

| Seq. | **Task 3: Three Supporting Facts** | $\alpha^1$ | $\alpha^2$ |
|------|-------------------------------------|-----------|-----------|
| 33 | Daniel took the football . | 0.46 | 0.01 |
| 39 | Sandra travelled to the bedroom . | 0.01 | 0.00 |
| 40 | Daniel moved to the bathroom . | 0.01 | 0.31 |
| 41 | Sandra got the milk . | 0.00 | 0.00 |
| 42 | Daniel travelled to the garden . | 0.04 | 0.42 |
| 46 | Daniel went to the hallway . | 0.00 | 0.25 |
| 50 | Daniel put down the apple . | 0.00 | 0.01 |
| 51 | Daniel put down the football there . | 0.38 | 0.00 |
| User input | Where was the football before the garden ? | | |
| Answer | Bathroom | | |
| Model answer | Bathroom | [Correct] | |

| Seq. | **Task 3: Three Supporting Facts** | $\alpha^1$ | $\alpha^2$ |
|------|-------------------------------------|-----------|-----------|
| 1 | Mary got the football . | 0.26 | 0.00 |
| 3 | Mary picked up the football . | 0.39 | 0.00 |
| 5 | Mary moved to the office . | 0.00 | 0.05 |
| 6 | Mary went to the hallway . | 0.00 | 0.05 |
| 11 | Mary travelled to the garden . | 0.00 | 0.21 |
| 12 | Mary travelled to the kitchen . | 0.00 | 0.29 |
| 14 | Mary moved to the office . | 0.00 | 0.25 |
| 16 | Mary went to the garden . | 0.00 | 0.05 |
| 22 | Mary discarded the football there . | 0.24 | 0.00 |
| User input | Where was the football before the garden ? | | |
| Answer | Office | | |
| Model answer | Kitchen | [Incorrect] | |

## 5 RESULTS AND DISCUSSION

### 5.1 BABI STORY-BASED QA

As we can see in table 2, RMN shows state-of-the-art result on bAbI story-based Question Answering dataset: 98.8% accuracy where only a single task with no catastrophic failure. It succeeded on task 17, 18, 19 where MemN2N, GMemN2N, and DMN are failed to solve, and on task 16 which DMN+, DNC, and EntNet scored high error rates.

Table 3 shows how our model solved several tasks. RMN's attention component $g_\theta^1$ and $g_\theta^2$ complement each other to identify the necessary facts to answer correctly. Sometimes both $g_\theta^1$ and $g_\theta^2$ concentrate on the same sentences which are all critical to answer the question, and sometimes $g_\theta^1$

---

[1] For a fair comparison, we report EntNet's result which was jointly trained on all tasks. It was written in the appendix of the paper.

[2] Our implementation. The result is different from what Santoro et al. (2017) mentioned, which is caused by the initialization.

finds a fact related to the given question and with this information $g_\theta^2$ chooses the key fact to answer. While trained jointly, RMN learns these different solutions for each task. For the task 3, the only failed task, attention component still functions well; it focuses sequentially on the supporting sentences. However, the reasoning component, $f_\phi$, had difficulty catching the word 'before'. We could easily figure out 'before' implies 'just before' the certain situation, whereas RMN confused its meaning. As shown in table 3c, our model found all previous locations before the garden. Still, it is remarkable that the simple MLP carried out all of these various roles.

## 5.2 BABI DIALOG

Table 4: Test error on bAbI dialog tasks [3]

| Task | Plain | | | | With Match | | | |
|---|---|---|---|---|---|---|---|---|
| | MemN2N | GMemN2N | RN[4] | RMN | MemN2N | GMemN2N | RN[4] | RMN |
| 1: Issuing API calls | 0.1 | **0.0** | **0.0** | **0.0** | **0.0** | **0.0** | **0.0** | **0.0** |
| 2: Updating API calls | **0.0** | **0.0** | 0.5 | **0.0** | 1.7 | **0.0** | **0.0** | **0.0** |
| 3: Displaying options | **25.1** | **25.1** | 26.6 | **25.1** | **25.1** | **25.1** | 27.1 | **25.1** |
| 4: Providing extra information | 40.5 | 42.8 | **0.0** | **0.0** | **0.0** | **0.0** | **0.0** | **0.0** |
| 5: Conducting full dialogs | 3.9 | 3.7 | 23.3 | **2.5** | 6.6 | 2.0 | 16.6 | **1.8** |
| Average error rates (%) | 13.9 | 14.3 | 10.1 | **5.5** | 6.7 | 5.4 | 8.7 | **5.4** |
| 1 (OOV): Issuing API calls | 27.7 | 17.6 | 17.8 | **16.8** | 3.5 | **0.0** | 1.5 | **0.0** |
| 2 (OOV): Updatating API calls | **21.1** | **21.1** | 23.2 | **21.1** | 5.5 | 5.8 | **0.0** | **0.0** |
| 3 (OOV): Displaying options | 25.6 | **24.7** | 27.2 | 24.9 | **24.8** | 24.9 | 29.8 | 25.1 |
| 4 (OOV): Providing extra information | 42.4 | 43.0 | **0.0** | **0.0** | **0.0** | **0.0** | **0.0** | **0.0** |
| 5 (OOV): Conducting full dialogs | 34.5 | **33.3** | 38.3 | 34.5 | 22.3 | 20.6 | 28.4 | 21.7 |
| Average error rates (%) | 30.3 | 27.9 | 21.3 | **19.5** | 11.2 | 10.3 | 12.0 | **9.4** |

The results in the Table 4 show that the RMN has the best results in any conditions. Without any match type, RN and RMN outperform previous memory-augmented models on both normal and OOV tasks. This is mainly attributed to the impressive result on task 4 which can be interpreted as an effect of MLP based output feature map.

To solve task 4, it is critical to understand the relation between 'phone number' of user input and 'r_phone' of previous dialog as shown in Table 8c. We assumed that inner product was not sufficient to capture their implicit similarity and performed an supporting experiment. We converted RMN's attention component to inner product based attention, and the results revealed the error rate increased to 11.3%.

For the task 3 and task 5 where the maximum length is especially longer than the others, RN performs worse than MemN2N, GMemN2N and RMN. The number of unnecessary object pairs created by the RN not only increases the processing time but also decreases the accuracy.

With the match type feature, all models other than RMN have significantly improved their performance except for task 3 compared to the plain condition. RMN was helped by the match type only on the OOV tasks and this implies RMN is able to find relation in the With Match condition for the normal tasks.

When we look at the OOV tasks more precisely, RMN failed to perform well on the OOV task 1 and 2 even though $g_\theta^1$ properly focused on the related object as shown in Table 8a. We state that this originated from the fact that the number of keywords in task 1 and 2 is bigger than that in task 4. In task 1 and 2, all four keywords (cuisine, location, number and price) must be correctly aligned from the supporting sentence in order to make the correct API call which is harder than task 4. Consider the example in Table 8a and Table 8c. Supporting sentence of task 4 have one keyword out of three words, whereas supporting sentences of task 1 and 2 consist of four keywords (cuisine, location, number and price) out of sixteen words.

Different from other tasks, RMN yields the same error rate 25.1% with MemN2N and GMemN2N on the task 3. The main goal of task 3 is to recommend restaurant from knowledge base in the order of rating. All failed cases are displaying restaurant where the user input is <silence>which is somewhat an ambiguous trigger to find the input relevant previous utterance. As shown in Table 8b, there are two different types of response to the same user input. One is to check whether all the

---

[3]We only compare models under the same conditions

[4]Our implementation.

required fields are given from the previous utterances and then ask user for the missing fields or send a "Ok let me look into some options for you." message. The other type is to recommend restaurant starting from the highest rating. All models show lack of ability to discriminate these two types of silences so that concluded to the same results. To verify our statement, we performed an additional experiment on task 3 and checked the performance gain (extra result is given in Table 10 of Appendix B).

## 5.3 MODEL ANALYSIS

**Effectiveness of the MLP-based output feature map**   The most important feature that distinguishes MemNN based models is the output feature map. Table 5 summarizes the experimental results for the bAbI story-based QA dataset when replacing the RMN's MLP-based output feature map with the idea of the previous models. inner product was used in MemN2N, inner product with gate was used in GMemN2N, and inner product and absolute difference with two embedding matrices was used in DMN and DMN+. From the Table 5, the more complex the output feature map, the better the overall performance. In this point of view, MLP is the effective output feature map.

Table 5: Test error of RMN on bAbI story-based QA dataset with different configurations

|  | inner product | inner product with gate | inner product and absolute difference with two embedding matrices | MLP |
|---|---|---|---|---|
| error rate | 29.4 | 25.9 | 11.2 | **1.2** |

**Performance of RN and RMN according to memory size**   Additional experiments were conducted with the bAbI story-based QA dataset to see how memory size affects both performance and training time of RN and RMN. Test errors with training time written in parentheses are summarized in Table 6.

When memory size is small, we could observe the data-effeciency of RN. It shows similar performance to RMN in less time. However, when the memory size increases, performance is significantly reduced compared to RMN, even though it has been learned for a longer time. It is even lower than itself when the memory size is 20. On the other hand, RMN maintains high performance even when the memory size increases.

Table 6: Test error and training time of RN and RMN on bAbI story-based QA dataset with different memory size

| memory size | RN | RMN |
|---|---|---|
| 20 | 2.0 (0.65 days) | 1.5 (1.46 days) |
| 130 | 9.8 (9.47 days) | **1.2** (4.94 days) |

**Effectiveness of the number of hops**   bAbI story based QA dataset differs in the number of supporting sentences by each task that need to be referenced to solve problems. For example, task 1, 2, and 3 require single, two, and three supporting facts, respectively. The result of the mean error rate for each task according to the number of hops is in Table 7.

Overall, the number of hops is correlated with the number of supporting sentences. In this respect, when the number of relations increases, RMN could reason across increasing the number of hops to 3, 4 or more.

Table 7: Test error of RMN on bAbI story-based QA dataset with different number of hops

|  | hop 1 | hop 2 | hop 3 |
|---|---|---|---|
| task 1 (1) [5] | **0.0** | **0.0** | 0.2 |
| task 2 (2) | 62.0 | **0.5** | 2.1 |
| task 3 (3) | 62.4 | 14.7 | **14.6** |
| task 10 (1) | **0.0** | **0.0** | 3.6 |

---

[5] Number in parentheses indicates the number of supporting sentences to solve the task

## 6    CONCLUSION

Our work, RMN, is a simple and powerful architecture that effectively handles text-based question answering tasks when large size of memory and high reasoning ability is required. Multiple access to the external memory to find out necessary information through a multi-hop approach is similar to most existing approaches. However, by using a MLP that can effectively deal with complex relatedness when searching for the right supporting sentences among a lot of sentences, RMN raised the state-of-the-art performance on the story-based QA and goal-oriented dialog dataset. When comparing RN which also used MLP to understand relations, RMN was more effective in the case of large memory.

Future work will apply RMN to image based reasoning task (e.g., CLEVR, DAQUAR, VQA etc.). To extract features from the image, VGG net (Simonyan & Zisserman, 2014) is used in convention and outputs 196 objects of 512 dimensional vectors which also require large sized memory. An important direction will be to find an appropriate way to focus sequentially on related object which was rather easy in text-based reasoning.

Table 8: Visualization of $\alpha$ on bAbI dialog task 1, 3, and 4 without match type

(a) Plain and OOV result of task 1

| Seq. | Locutor | Task 1: Issuing API calls | $\alpha^1$ |
|---|---|---|---|
| 1 | user | Good morning | 0.01 |
| 2 | bot | Hello what can i help you with today | 0.01 |
| 3 | user | Can you make restaurant reservation for eight people in cheap price range with british cuisine | 0.23 |
| 4 | bot | I'm on it | 0.00 |
| 5 | user | <silence> | 0.14 |
| 6 | bot | Where should it be | 0.01 |
| 7 | user | In madrid | 0.25 |
| 8 | bot | Ok let me look into some options for you | 0.18 |
| User input | | <silence> | |
| Answer | | API_call british madrid eight cheap | |
| Model answer | | API_call british madrid eight cheap           [Correct] | |
| Seq. | Locutor | Task 1 (OOV): Issuing API calls | $\alpha^1$ |
| 1 | user | Hi | 0.01 |
| 2 | bot | Hello what can i help you with today | 0.01 |
| 3 | user | May I have table with cantonese cuisine in cheap price range in seoul for two people | 0.49 |
| 4 | bot | I'm on it | 0.10 |
| 5 | user | <silence> | 0.01 |
| 6 | bot | Ok let me look into some options for you | 0.27 |
| User input | | <silence> | |
| Answer | | API_call cantonese seoul two cheap | |
| Model answer | | API_call italian paris two cheap           [Incorrect] | |

(b) Same user input with different goal in task 3

| Seq. | Locutor | Task 3: Displaying options | $\alpha^1$ |
|---|---|---|---|
| 7 | user | resto_1 r_rating 1 | 0.01 |
| 14 | user | resto_8 r_rating 8 | 0.01 |
| 21 | user | resto_3 r_rating 3 | 0.01 |
| 22 | user | Hello | 0.01 |
| 23 | bot | Hello what can i help you with today | 0.01 |
| 24 | user | Can you book table with french food for two people in madrid in moderate price range | 0.21 |
| 25 | bot | I'm on it | 0.01 |
| User input | | <silence> | |
| Answer | | Where should it be? | |
| Model answer | | Where should it be?           [Correct] | |
| Seq. | Locutor | Task 3: Displaying options | $\alpha^1$ |
| 7 | user | resto_1 r_rating 1 | 0.00 |
| 14 | user | resto_8 r_rating 8 | 0.00 |
| 21 | user | resto_3 r_rating 3 | 0.00 |
| 24 | user | Can you book table with french food for two people in madrid in moderate price range | 0.14 |
| 25 | bot | I'm on it | 0.00 |
| 26 | user | <silence> | 0.09 |
| 27 | bot | Ok let me look into some options for you | 0.15 |
| 28 | user | <silence> | 0.10 |
| 29 | bot | What do you think of this option : resto_8 | 0.08 |
| 30 | user | No this does not work for me | 0.06 |
| 31 | bot | Sure let me find other option for you | 0.08 |
| User input | | <silence> | |
| Answer | | What do you think of this option: resto_3 | |
| Model answer | | What do you think of this option: resto_7           [Incorrect] | |

(c) Plain and OOV result of task 4

| Seq. | Locutor | Task 4: Providing extra information | $\alpha^1$ |
|---|---|---|---|
| 1 | user | resto_3 r_phone resto_3_phone | 0.77 |
| 3 | user | resto_3 r_address resto_3_address | 0.01 |
| 4 | user | resto_3 r_location london | 0.01 |
| 6 | user | resto_3 r_price cheap | 0.01 |
| 10 | user | I'd like to book table at resto_3 | 0.02 |
| 11 | bot | Great let me do reservation | 0.03 |
| User input | | What is phone number of restaurant? | |
| Answer | | Here it is: resto_3_phone | |
| Model answer | | Here it is: resto_3_phone           [Correct] | |
| Seq. | Locutor | Task 4 (OOV): Providing extra information | $\alpha^1$ |
| 1 | user | resto_1 r_phone resto_1_phone | 0.79 |
| 3 | user | resto_1 r_address resto_1_address | 0.01 |
| 4 | user | resto_1 r_location hanoi | 0.01 |
| 6 | user | resto_1 r_price expensive | 0.01 |
| 10 | user | Can you make restaurant reservation at resto_1 | 0.01 |
| 11 | bot | Great let me do reservation | 0.04 |
| User input | | What is phone number of restaurant? | |
| Answer | | Here it is: resto_1_phone | |
| Model answer | | Here it is: resto_1_phone           [Correct] | |

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

## A  MODEL DETAILS

Table 9: Hyperparameters of Relation Memory Networks on bAbI dialog tasks

| Task | Story and Question Embedding | Word-lookup Embedding Dim | Hop | $g_\theta$ | $f_\phi$ | Activation | Use Batch Norm |
|---|---|---|---|---|---|---|---|
| 1 | sum | 128 | 1 | 2048, 2048, 1 | 2048, 2048, 4212 | tanh | True |
| 2 | sum | 128 | 1 | 1024, 1024, 1 | 1024, 1024, 4212 | tanh | True |
| 3 | sum | 128 | 1 | 1024, 1024, 1024, 1 | 1024, 1024, 1024, 4212 | tanh | True |
| 4 | concatenation | 50 | 1 | 1024, 1024, 1 | 1024, 1024, 4212 | tanh | True |
| 5 | concatenation | 64 | 2 | 4096, 4096, 1 | 4096, 4096, 4212 | tanh | True |

## B  ADDITIONAL RESULTS

Table 10: Visualization of $\alpha^1$ and $\alpha^2$ on user input revised bAbI dialog task 3 without match type

| Seq. | Locutor | Task 3: Displaying options | $\alpha^1$ | $\alpha^2$ |
|---|---|---|---|---|
| 1 | user | resto_8 r_phone resto_8_phone | 0.00 | 0.01 |
| 2 | user | resto_8 r_cuisine french | 0.01 | 0.01 |
| 3 | user | resto_8 r_address resto_8_address | 0.01 | 0.01 |
| 4 | user | resto_8 r_location madrid | 0.01 | 0.01 |
| 5 | user | resto_8 r_number two | 0.00 | 0.01 |
| 6 | user | resto_8 r_price moderate | 0.00 | 0.01 |
| 7 | user | resto_8 r_rating 8 | 0.01 | 0.39 |
| 8 | user | resto_3 r_phone resto_3_phone | 0.01 | 0.01 |
| 9 | user | resto_3 r_cuisine french | 0.00 | 0.01 |
| 10 | user | resto_3 r_address resto_3_address | 0.00 | 0.01 |
| 11 | user | resto_3 r_location madrid | 0.00 | 0.01 |
| 12 | user | resto_3 r_number two | 0.00 | 0.01 |
| 13 | user | resto_3 r_price moderate | 0.00 | 0.01 |
| 14 | user | resto_3 r_rating 3 | 0.00 | 0.01 |
| 15 | user | resto_1 r_phone resto_1_phone | 0.01 | 0.01 |
| 16 | user | resto_1 r_cuisine french | 0.01 | 0.01 |
| 17 | user | resto_1 r_address resto_1_address | 0.01 | 0.01 |
| 18 | user | resto_1 r_location madrid | 0.01 | 0.01 |
| 19 | user | resto_1 r_number two | 0.00 | 0.01 |
| 20 | user | resto_1 r_price moderate | 0.00 | 0.01 |
| 21 | user | resto_1 r_rating 1 | 0.00 | 0.01 |
| 22 | user | Good morning | 0.01 | 0.01 |
| 23 | bot | Hello what can I help you with today? | 0.12 | 0.01 |
| 24 | user | May I have table for two with french food? | 0.08 | 0.02 |
| 25 | bot | I'm on it | 0.03 | 0.01 |
| 26 | user | <silence> | 0.01 | 0.01 |
| 27 | bot | Where should it be | 0.09 | 0.01 |
| 28 | user | In madrid | 0.01 | 0.01 |
| 29 | bot | Which price range are looking for? | 0.09 | 0.01 |
| 30 | user | In moderate price range please | 0.01 | 0.01 |
| 31 | bot | Ok let me look into some options for you | 0.24 | 0.01 |
| User input | | <silence><silence> | | |
| Answer | | What do you think of this option: resto_8 | | |
| Model answer | | What do you think of this option: resto_8 | **[correct]** | |

We modify the user input from <silence> to <silence><silence> when looking for restaurant recommendations. This makes model to distinguish two different situations whether to ask for additional fields or to recommend restaurant.

