# OpenReview forum: "Finding ReMO (Related Memory Object): A Simple neural architecture for Text based Reasoning"
_ICLR.cc/2018/Conference — Reject_

### Official Review · AnonReviewer2 · 2017-11-27
**Not sure what is novel**

**Rating:** 4
**Confidence:** 4

**Review:**

The paper proposes to address the quadratic memory/time requirement of Relation Network (RN) by sequentially attending (via multiple layers) on objects and gating the object vectors with the attention weights of each layer. The proposed model obtains state of the art in bAbI story-based QA and bAbI dialog task.

Pros:
- The model achieves the state of the art in bAbI QA and dialog. I think this is a significant achievement given the simplicity of the model.
- The paper is clearly written.

Cons:
- I am not sure what is novel in the proposed model. While the authors use notations used in Relation Network (e.g. 'g'), I don't see any relevance to Relation Network. Rather, this exactly resembles End-to-end memory network (MemN2N) and GMemN2N. Please tell me if I am missing something, but I am not sure of the contribution of the paper. Of course, I notice that there are small architectural differences, but if these are responsible for the improvements, I believe the authors should have conducted ablation study or qualitative analysis that show that the small tweaks are meaningful.

Question:
- What is the exact contribution of the paper with respect to MemN2N and GMemN2N?

---

> ### Author Response · Authors · 2017-12-24
> **Response to Reviewer2**
>
> Thank you for your review. You raise a good point that help us clarify and improve the paper.
> It took us quite a long time to do some additional experiments on the point you pointed out.
>
>
> "While the authors use notations used in Relation Network (e.g. 'g'), I don't see any relevance to Relation Network. Rather, this exactly resembles End-to-end memory network (MemN2N) and GMemN2N."
>
> --- Our response))
> Since the proposed model (RMN) follows the framework of Memory Network, it has a similar structure to MemN2N and GMemN2N which are also Memory Network based models.
> The reason for mentioning RN is that the MLP-based attention mechanism of the RMN is motivated by the RN's g.
> When analyzing the structure of the RN from the viewpoint of the memory network, it can be seen that the g of the RN plays the same role as the output feature map which takes charge of the attention mechanism among the components of the memory network.
> We re-described this on the paper to clarify and added a table comparing MemN2N, RN, and RMN.
>
> "What is the exact contribution of the paper with respect to MemN2N and GMemN2N?"
>
> --- Our response))
> We think it is the most critical question, and I acknowledge that I have written the paper unclear.
> We have revised the paper so that we can answer this question.
>
> For your question, I would like to say that all models based on Memory Network (MemN2N, GMemN2N etc) show small differences.
> For example, GMemN2N only added gate operation compared to MemN2N.
> In this respect, our model’s contribution is the MLP-based attention mechanism.
> We thought that the reasoning ability of the model depends on how well the relevant sentence are found in the memory.
> The performance of MemN2N, GMemN2N, and DMN + was improved in the order of the attention mechanism becoming complex.
> Therefore, RMN is designed to have an overall simple structure while having MLP-based attention mechanism to catch the complicate relation.
> To validate this effect, we added model analysis as a subsection to the discussion and conducted an ablation study to compare the results according to the approach of the attention mechanism.
> Also, we designed an updating component that fits well on our attention component.

---

### Official Review · AnonReviewer1 · 2017-11-27
**Finding ReMO review**

**Rating:** 4
**Confidence:** 4

**Review:**

This paper introduces Related Memory Network (RMN), an improvement over Relationship Networks (RN). RMN avoids growing the relationship time complexity as suffered by RN (Santoro et. Al 2017). RMN reduces the complexity to linear time for the bAbi dataset. RN constructs pair-wise interactions between objects in RN to solve complex tasks such as transitive reasoning. RMN instead uses a multi-hop attention over objects followed by an MLP to learn relationships in linear time.

Comments for the author:

The paper addresses an important problem since understanding object interactions are crucial for reasoning. However, how widespread is this problem across other models or are you simply addressing a point problem for RN? For example, Entnet is able to reason as the input is fed in and the decoding costs are low. Likewise, other graph-based networks (which although may require strong supervision) are able to decode quite cheaply.

The relationship network considers all pair-wise interactions that are replaced by a two-hop attention mechanism (and an MLP). It would not be fair to claim superiority over RN since you only evaluate on bABi while RN also demonstrated results on other tasks. For more complex tasks (even over just text), it is necessary to show that you outperform RN w/o considering all objects in a pairwise fashion. More specifically, RN uses an MLP over pair-wise interactions, does that allow it to model more complex interactions than just selecting two hops to generate attention weights. Showing results with multiple hops (1,2,..) would be useful here.

More details are needed about Figure 3. Is this on bAbi as well? How did you generate these stories with so many sentences? Another clarification is the bAbi performance over Entnet which claims to solve all tasks. Your results show 4 failed tasks, is this your reproduction of Entnet?

Finally, what are the savings from reducing this time complexity? Some wall clock time results or FLOPs of train/test time should be provided since you use multiple hops.

Overall, this paper feels like a small improvement over RN. Without experiments over other datasets and wall clock time results, it is hard to appreciate the significance of this improvement. One direction to strengthen this paper is to examine if RMN can do better than pair-wise interactions (and other baselines) for more complex reasoning tasks.

---

> ### Author Response · Authors · 2017-12-24
> **Response to Reviewer1**
>
> We thank the reviewer for the points of clarification and correction.
> We have modified the paper to address these points, and include detailed answers about each question below.
>
> "how widespread is this problem across other models or are you simply addressing a point problem for RN?”
>
> — Our response ))
> It seems to have asked this question because the issue of the submitted paper was unclear (We revised it to be clear) .
> In fact, our paper suggests a new framework suitable for text-based reasoning rather than solving the problems of RN.
> While suggesting RMN, it shows the possibility of replacing the pair-wise interaction of RN.
>
>
> "It would not be fair to claim superiority over RN since you only evaluate on bAbI while RN also demonstrated results on other tasks. For more complex tasks (even over just text), it is necessary to show that you outperform RN w/o considering all objects in a pairwise fashion.”
>
> — Our response ))
> As RMN is a new framework for text-based reasoning, we didn’t perform additional experiment over text, like image.
> Rather, we conducted experiments on bAbI dialog-based QA dataset for rich discussion.
>
>
> "RN uses an MLP over pair-wise interactions, does that allow it to model more complex interactions than just selecting two hops to generate attention weights. Showing results with multiple hops (1,2,..) would be useful here."
>
> — Our response )
> I’m not sure that I understood you comment as you intended to.
> Please let me know if my response is not enough to this question.
> You said that RN is allowed to model more complex interactions than just two hops, however, at least in text-based QA dataset, it is revealed not always true.
> If we take a closer look at our model, RMN is also able to model complex interactions.
> When hop 1 result, r1, is concatenated with updated memory, it is similar to the object pair that RN is dealing with.
> Therefore RMN is able to handle complicate interaction as much as RN is.
>
> Also we add the result with multiple hops and it reveals that the number of hops is correlated with the number of relation.
>
>
> "More details are needed about Figure 3. Is this on bAbi as well? How did you generate these stories with so many sentences?"
>
> — Our response ))
> Sorry for the unclear description about Figure 3.
> It was tested on the bAbI story based QA dataset because it has 320 sentences on one story at maximum.
> Traditionally, Memory Network based models test their performance on 130 sentences at maximum for task 3 and 70 sentences at maximum for the others.
> RN’s experiment was a special case that tested on 20 sentences.
> However our revised paper does not include this figure anymore.
>
>
> "Some wall clock time results or FLOPs of train/test time should be provided since you use multiple hops. what are the savings from reducing this time complexity?"
>
> — Our response ))
> We changed our comparison with RN to model accuracy and training time when memory size is large and small.
> Our reduction in the time complexity leads to shorter training time than RN when memory size is large.
> In addition, when memory is large, RN’s reasoning ability is decreased while RMN still shows good reasoning ability.
>
>
> "Another clarification is the bAbI performance over Entnet which claims to solve all tasks.”
>
> — Our response ))
> Including most of other models, such as MemN2N, DMN+, and RN, we also conducted experiment in jointly rather than task-wise. On the other hand, EntNet's results which claims to solve all tasks, are the results of task-wise condition. For fair comparison, we used the jointly trained results of the EntNet which is described in the Appendix of EntNet paper.
> To make clear, I add this to footnote.

---

### Official Review · AnonReviewer3 · 2017-11-29

**Rating:** 4
**Confidence:** 4

**Review:**

This paper proposes an alternative to the relation network architecture whose computational complexity is linear in the number of objects present in the input. The model achieves good results on bAbI compared to memory networks and the relation network model. From what I understood, it works by computing a weighted average of sentence representations in the input story where the attention weights are the output of an MLP whose input is just a sentence and question (not two sentences and a question). This average is then fed to a softmax layer for answer prediction. I found it difficult to understand how the model is related to relation networks, since it no longer scores every combination of objects (or, in the case of bAbI, sentences), which is the fundamental idea behind relation networks. Why is the approach not evaluated on CLEVR, in which the interaction between two objects is perhaps more critical (and was the main result of the original relation networks paper)? The fact that the model works well on bAbI despite its simplicity is interesting, but it feels like the paper is framed to suggest that object-object interactions are not necessary to explicitly model, which I can't agree with based solely on bAbI experiments. I'd encourage the authors to do a more detailed experimental study with more tasks, but I can't recommend this paper's acceptance in its current form.

other questions / comments:
- "we use MLP to produce the attention weight without any extrinsic computation between the input sentence and the question." isn't this statement false because the attention computation takes as input the concatenation of the question and sentence representation?
- writing could be cleaned up for spelling / grammar (e.g., "last 70 stories" instead of "last 70 sentences"), currently the paper is very hard to read and it took me a while to understand the model

---

> ### Author Response · Authors · 2017-12-25
> **Response to Reviewer3**
>
> Thank you for your review. Based on the points you mentioned, I revised the paper.
> Below is your review and my answer to that.
>
> "I found it difficult to understand how the model is related to relation networks, since it no longer scores every combination of objects (or, in the case of bAbI, sentences), which is the fundamental idea behind relation networks.”
> — Our response ))
> In the past, this point has not been clarified, so we have revised paper to emphasize on how RMN is related to relation network.
> Our model is a new text-based reasoning model based on the Memory Network framework.
> In text-based reasoning, the most important thing is to select supporting sentences from large memory, which is performed through attention mechanism.
> We found that the performance increases with more complex attention mechanisms.
> As RN is one of the models that reasons well, we analyzed RN from the perspective of Memory Network.
> We found out that the g of RN examines the relatedness of object pair and question very well.
> Motivated from it, we also used the MLP to focus on the supporting sentences examined from the relatedness of object and question in the memory network framework.
> As a result, we were motivated by the fact that MLP was effective to examine the relatedness rather than the modeling structure of the RN that use object pair combination.
>
>
> "Why is the approach not evaluated on CLEVR, in which the interaction between two objects is perhaps more critical (and was the main result of the original relation networks paper)?”
> — Our response ))
> This is because our model is a new model for text-based reasoning based on Memory Network.
> I also thought about evaluating our model on images.
> However, since it is Memory Network based reasoning model, I wanted to verify the performance of the model for text, first.
>
>
> "I'd encourage the authors to do a more detailed experimental study with more tasks."
> — Our response ))
> We added the experimental results of the RN to the bAbI dialog-based dataset and discussed it on the paper.
> In addition, we compared training time and performance of RN to our model in a large memory condition.
>
>
>
> “ "we use MLP to produce the attention weight without any extrinsic computation between the input sentence and the question." isn't this statement false because the attention computation takes as input the concatenation of the question and sentence representation?"
> — Our response ))
> Extrinsic attention computation refers to inner product and absolute difference performed by MemN2N, GMemN2N, DMN+ when relatedness of question and sentence is calculated.
> On the other hand, there is no computation conducted in RN and RMN because they use simple concatenation.
>
>
> "writing could be cleaned up for spelling / grammar (e.g., "last 70 stories" instead of "last 70 sentences”)"
> — Our response ))
> I have reviewed a number of times but have not been able to catch them.
> Thank you for pointing out and I removed such content from the new paper.

---

### Author Response · Authors · 2017-12-24
**Revised paper**

We’ve uploaded a new version of the paper that addresses much of the reviewers’ comments and questions.
Also we included new experiments that throws a light on the modeling effect of RMN.
The additional experiments are as follows.
1) RN's result on bAbI dialog based QA dataset.
2) RN's result on bAbI story based QA dataset.
3) Ablation study on RMN where attention mechanism is changed.
4) Result of RN and RMN where memory size is varied.
5) Result of RMN according to the number of hops

The results show that RMN is better at text-based reasoning compared to MemN2N, GMemN2N, and other Memory Network based models.
In addition, when compared to Relation Network, RMN's strong reasoning ability is revealed when memory size in large.
We also found the correlation between the number of hops and the number of relations.

Comments from reviewers have helped to clarify the paper, and we have revised it to talk more clearly about the contribution of our model.

---

### Decision · Program_Chairs · 2018-01-29
**ICLR 2018 Conference Acceptance Decision**

**Decision:**

Reject

**Comment:**

The contribution of this paper basically consists of using MLPs in the attention mechanism of end-2-end memory networks. Though it leads to some improvements on bAbI (which may not be so surprising - MLP attention has been shown preferable in certain scenarious), it does not seem to be a sufficient contribution. The motivation is also confusing - the work is not really that related to relation networks, which were specifically designed to deal with situations where *relations* between objects matter. The proposed architecture does not model relations.

+  improvement on bAbI over the baselines
-  limited novelty (MLP attention is fairly standard)
-  the presentation of the idea is confusing (if the claim is about relations -> other datasets need to be considered)

There is a consensus between reviewers.